# Impact of Fenugreek on Milk Production in Rodent Models of Lactation Challenge

**DOI:** 10.3390/nu11112571

**Published:** 2019-10-24

**Authors:** Thomas Sevrin, Marie-Cécile Alexandre-Gouabau, Blandine Castellano, Audrey Aguesse, Khadija Ouguerram, Patrick Ngyuen, Dominique Darmaun, Clair-Yves Boquien

**Affiliations:** 1Laboratoire FRANCE Bébé Nutrition, 53000 Laval, France; thomas.sevrin@etu.univ-nantes.fr; 2Nantes Université, INRA, UMR 1280 PhAN, CRNH-Ouest, IMAD, F-44000 Nantes, France; Marie-Cecile.Alexandre-Gouabau@univ-nantes.fr (M.-C.A.-G.); Blandine.Castellano@univ-nantes.fr (B.C.); Audrey.Aguesse@univ-nantes.fr (A.A.); Khadija.Ouguerram@univ-nantes.fr (K.O.); Dominique.Darmaun@chu-nantes.fr (D.D.); 3ONIRIS, NP3, CRNH-Ouest, F-44000 Nantes, France; patrick.nguyen@oniris-nantes.fr; 4CHU Nantes, F-44000 Nantes, France

**Keywords:** fenugreek, milk flow, milk composition, litter size, maternal protein restriction, plasma metabolic parameters

## Abstract

Fenugreek, a herbal remedy, has long been used as galactologue to help mothers likely to stop breastfeeding because of perceived insufficient milk production. However, few studies highlight the efficacy of fenugreek in enhancing milk production. The aims of our study were to determine whether fenugreek increased milk yield in rodent models of lactation challenge and if so, to verify the lack of adverse effects on dam and offspring metabolism. Two lactation challenges were tested: increased litter size to 12 pups in dams fed a 20% protein diet and perinatal restriction to an 8% protein diet with eight pups’ litter, with or without 1 g.kg^−1^.day^−1^ dietary supplementation of fenugreek, compared to control dams fed 20% protein diet with eight pups’ litters. Milk flow was measured by the deuterium oxide enrichment method, and milk composition was assessed. Lipid and glucose metabolism parameters were assessed in dam and offspring plasmas. Fenugreek increased milk production by 16% in the litter size increase challenge, resulting in an 11% increase in pup growth without deleterious effect on dam-litter metabolism. Fenugreek had no effect in the maternal protein restriction challenge. These results suggest a galactologue effect of fenugreek when mothers have no physiological difficulties in producing milk.

## 1. Introduction

The World Health Organisation recommends exclusive breastfeeding for infants up to six months of age, based on the clear health benefits of breastfeeding on mother-infant dyad [1]. Indeed, there is a consensus regarding the association of breastfeeding with a reduced risk of respiratory and gastro-intestinal infections during the first year of life. Infants, who were breastfed for longer periods, may also have a lower risk of developing obesity and type II diabetes at adulthood [2,3]. For mothers, breastfeeding could limit the risk of developing ovarian cancer and type II diabetes [4,5]. Despite these benefits, breastfeeding prevalence remains relatively low, particularly in several high-income countries in North America and Europe, where only 40% of mothers breastfeed six months after delivery [6]. Exclusive breastfeeding rate is about 60% at four months in Scandinavian countries, 35% in the Netherlands, 16% in the UK [7], and 10% in France [7,8]. 

Early cessation of breastfeeding clearly is multifactorial [9]. One of the main factors is the maternal perception of insufficient milk secretion to quell infant’s hunger or support infant growth and leads to early cessation of breastfeeding in 35% of cases [9,10,11]. Perception of insufficient milk production is a complex, multifactorial issue that can have biological, social, or psychological determinants, and it often remains unclear whether the low milk secretion is real or only perceived [9]. Indeed, as a result of the lack of an objective marker of insufficient milk production and the importance of maternal psychology in breastfeeding duration, perceived milk insufficiency is probably much more common than true insufficient production [9,10,11]. True insufficient milk secretion can result from many causes, ranging from inability to lactate due to breast abnormalities or endocrine disorder (5% of women) to difficulties in breastfeeding management, maternal stress and anxiety, or early food diversification in the infant. Although maternal milk production can be often increased through psychological support or maternal breastfeeding education [9,10,12], many healthy mothers are eager to enhance their milk production through various nutritional supplements.

Several drugs and herbal preparations have traditionally been prescribed as galactologues: i.e., substances that promote initiation or increase of lactation [13]. Drugs like domperidone® or metoclopramide® carry the risk of adverse side effects such as arrhythmia or hypothyroidism in mother-children dyad [12]. That is why herbal galactologues like fennel, anise, barley, milk thistle, or garlic are becoming more and more popular for increasing lactation. Among these herbal compounds, fenugreek probably is the most widely consumed [12,13,14]. Fenugreek has been used since antiquity in traditional Persian, Chinese, and Egyptian medicine for its range of therapeutic effects. It is now increasingly consumed in Western countries for its presumed protective effects against diabetes, atherosclerosis, inflammation, and hypertriglyceridemia, as well as its putative role as a galactologue [12,15] due, in part, to trigonelline, one of its main active ingredients [15]. There is, however, little evidence for its effectiveness on milk yield [13]. Whereas a positive effect of fenugreek on milk production was observed in various mammals, such as rabbit [16], buffalo [17], goat [18], or ewe [19]; the wide range of doses tested (from 180 mg.kg-1.d-1 [17] to 2.1 g.kg-1.d-1 [18]) led to large discrepancies in the reported effect on milk production (ranging from a 10% increase [16] to a 110% [18] increase), which makes it difficult to determine the effective dose of fenugreek. Moreover, other studies failed to demonstrate an effect in rabbits [20] and goats [21]. In these studies, milk production was measured either by the weight–suckle–weight method (in rabbit), or by milking (in larger animals). The latter methods do not, however, directly assess milk production in response to physiological suckling by the pups. Moreover, these studies evaluated the effectiveness of fenugreek to increase milk production with the aim of productivity, under otherwise optimal conditions of lactation, as opposed to mothers challenged by their perception of insufficient milk production. Finally, none of these studies evaluated the metabolic status of mother-infant dyad following fenugreek supplementation. Thus, the true efficacy of fenugreek on milk production in an animal model submitted to a breastfeeding challenge remains to be ascertained.

To that purpose, the current study used the deuterium oxide enrichment method [22], which meets all criteria to adequately assess breast milk supply even in small animals and, in turn, evaluate the putative galactologue effect of substances such as fenugreek. Indeed, it has been shown to be precise and to provide a smoothed value of several days milk production obtained under physiological conditions of lactation [22]. Moreover, in order to simulate conditions where mothers are unable to feed their own pups optimally, two classical models of lactation challenges were used. The first one mimics conditions in which mothers can adapt their milk production but still fail to adequately cover the litters’ needs, leading to suboptimal growth of the offspring. This model, obtained by increasing litter size through pup adoption, is known to induce extra-uterine growth restriction (EUGR) in the offspring due to a reduced milk intake in individual pups due to their larger number [23,24]. The second model mimics conditions in which mothers are truly unable to produce sufficient milk due to a perinatal restriction in dietary protein intake when fed a diet containing 8% protein instead of 20% in the standard diet. This model is also notably known to induce EUGR [25] due to a 34% drop in milk production [22].

The specific aims of this study were: (a) to verify the ability of our stable isotope method to detect changes in milk production in two rodent models of lactation challenge; (b) to test the galactologue effect of fenugreek on milk production and composition in these models of lactation challenge; and (c) to explore the effect of fenugreek on maternal metabolism during the lactation period and the short and long-term metabolic outcome in the offspring.

## 2. Materials and Methods

### 2.1. Animal Experiment

#### Housing and Diets

The experimental protocol was approved by the Animal Ethics committee and the French Ministry of Research (protocol APAFIS 2018121018129789). Pregnant Sprague-Dawley rats were purchased from Janvier Labs (Le Genest-Saint-Isle, France) at gestational day one (G1). They were housed individually in cages with wood chips located on ventilated racks kept at a constant temperature of 22 ± 1 °C and at a relative humidity of 50% ± 3%. Cages were placed in a room with a fixed 12 h light–dark cycle (light from 7:00 a.m. to 7:00 p.m.). Pregnant rats had access to water and food ad libitum. 

During gestation, dams received a standard normal protein diet based on AIN-93G diet [26] with 20 g protein per 100 g of food (NP diet) or an isoenergetic, low-protein diet with 8 g protein per 100 g of food (LP diet).

During lactation, dams received experimental diets based on NP and LP diets and supplemented with a dry water extract of fenugreek seeds (Plantex, Sainte-Geneviève-des-Bois, France) named NPF and LPF diets, respectively, in the following. Fenugreek, as an oxytocic substance [13], was provided to dams only during the lactation period. The amount of fenugreek in NPF and LPF diets was calculated to reach a consumption of 1 g.kg body weight^−1^.day^−1^ in rat as an equivalent as the traditionally recommended [12,13] 6 g daily dose in women weighing 60 kg and assuming that metabolic rate per unit of body weight is 10 fold higher in rats than in humans [27]. The four diets (i.e., NP, LP, NPF, and LPF) were manufactured by the “Unité de Préparation des Aliments Expérimentaux” (INRA-UPAE, Jouy-en-Josas, France). The composition and energy of each diet is provided in Appendix A.

After weaning, offspring were fed ad libitum with a standard growth diet A03 (SAFE, Augy, France).

### 2.2. Experimental Design

On the first day of gestation (G1), seventy-two female rats (i.e., 4 consecutive series of 18 animals), were randomly assigned to be fed experimental diets during gestation: 52 females received the NP diet and 20 females received the LP diet. Delivery occurred at the 21st day of gestation that was considered as day 0 of lactation (L0). At birth, pups born from protein-restricted dams (LP dams) were discarded and killed to avoid the bias due to intrauterine growth restriction. Only pups born from control dams (NP dams) were randomly adopted by NP or LP dams, taking care to balance pup’s birth weights between groups. The litter size was adjusted to either 8 or 12 pups per dam with a female/male ratio of 1/1, as described in Figure 1. 

Five groups were defined (47 dams). In the NP:8 group (our reference group, *n* = 8), dams suckled 8 pups per litter and were fed the NP diet. The NP:12 group (*n* = 11) corresponded to the first model of lactation challenge by increasing litter size (12 pups per litter) with dams under the NP diet. In the LP:8 group (*n* = 8), dams suckled 8 pups per litter and were fed the LP diet. The LP:8 group corresponded to the second model of lactation challenge with a known decreased milk flow due to perinatal protein restriction [22,25]. In the two other groups NPF:12 (12 suckled pups per litter, *n* = 11) and LPF:8 (8 suckled pups per litter, *n* = 9), dams were respectively fed an NP and LP diet supplemented with fenugreek.

At weaning (L20), 4 pups (2 males and 2 female) per litter were killed and perirenal, subcutaneous and brown adipose tissues were collected and weighed. Two other males and two females per litter were weaned and placed with 3 animals per cage with ad libitum access to food and water until post-natal day 75 (PND75), and then killed. Liver, gastrocnemius muscle, perirenal, subcutaneous, and visceral adipose tissues were removed from PND75 offspring and weighed.

At L21, mothers were fasted for 4h and then sacrificed. Liver, left inguinal mammary gland, perirenal, subcutaneous, and visceral adipose tissues were removed, weighed, sampled, and immediately frozen in liquid nitrogen before storage at −80 °C.

Sacrifice was performed by intracardiac injection of 0.5 mL Exagon® (Richter pharma, Wels, Austria) when animal tissues were collected, whereas supernumerary dams or supernumerary weaned pups were killed by carbon dioxide anaesthesia or by decapitation for pups at birth.

During lactation, dams’ weight, food and water consumption, and both male and female litters’ weight were recorded every two days from L0 to L11, and every day from L11 to L21. For dams, weight loss (in g) during the lactation period was calculated by subtracting weight at L0 (delivery) from the daily weight. For male and female offspring, pups’ mean weight (expressed as g) was obtained by dividing litter weight by the number of pups. Weight gain (in g) was calculated by subtracting birth weight (L0) from the daily weight. Daily growth rate (in g.day^−1^) was calculated by subtracting weight at day d-1 from the weight at day d. Food and water relative intakes (in g.kg^−1^.day^−1^) were obtained by dividing daily intakes by daily weight. 

### 2.3. Biological Samples Collection

Milk samples were collected from lactating dams at L18, as previously described [28]. Briefly, dams were separated from their pups and received an intraperitoneal injection of oxytocin (1 unit of Syntocinon®; Sigma-Tau, Ivry-sur-Seine, France) to stimulate milk ejection. After 20 min, dams were anaesthetized with 4% of isoflurane, and by applying manual pressure to nipples, about 200 to 400 µL of milk were collected before storage at −20 °C.

Blood samples were collected before animal sacrifice by an intra-cardiac puncture in EDTA-tubes (Pfizer-Centravet, Plancoët, France). Otherwise, blood samples were collected on alert dams by a tail snip in EDTA-tubes. Blood samples were centrifuged at 1132 g for 15 min at 4 °C, and plasma was collected in Eppendorf before storage at −20 °C until analysis. 

For urine, pups were first separated from the mother for 30 min to avoid urine loss by maternal stimulation. Urine samples were then collected from pups by stimulation of lower bellies with an iced cotton bud and pooled for the male and female litter.

### 2.4. Milk Flow Measurement by Water Turnover Method

The water turnover method was used as previously described [22]. Briefly, at L8, plasma and urine samples were collected prior to maternal deuterated water (D_2_O) injection to determine baseline body D_2_O abundance in dams and litter, respectively. At L11, when lactation was well established, following 4% isoflurane anaesthesia, mothers received an intravenous tail injection of 4.95 ± 0.13 g.kg^−1^ D_2_O (99.9 mole % D_2_-enrichment) (Sigma-Aldrich, Lyon, France). 

At 2 h, 24 h, 48 h, 72 h, and 96 h following D_2_O injection, dams’ blood samples (300 µL) were collected by a tail snip. At 24 h, 48 h, 72 h, 96 h, and 168 h following D_2_O-injection, a pool of urine (about 300 µL) was collected from both male and female pups of each litter. The D_2_O enrichment of both plasma and urine samples was measured using the Fourier Transform infrared spectrophotometer Alpha II® (Brucker, Rheinstetten, Germany).

Milk flow calculation, using water turnover method, has been previously described [22] and was refined in order to take into account both intra-litter sexual dimorphism and litter size differences between experimental groups. Regarding sexual dimorphism, instead of a bi-compartmental model (dam-litter), a four-compartment model (Figure 2) was used, in which 2 single-compartments corresponded to the turnover of total body water (TBW) of both male (3) and female (4) pups in the same litter, and the other 2 single-compartments corresponded to the turnover of TBW of their own dam (1 and 2, with 1 = 2). Each mother compartment was related by water flow to one pup compartment (male or female). Absolute production rates are represented in Figure 2: R10, R20 (R20 = R10), R30, and R40 are the inputs into the body of dam and its male and female litter, respectively, arising from water drinking and non-dietary water as metabolic water production; R01, R02, R03, and R04 are the outputs of water by transpiration, urine or faeces of dam and its male and female litter, respectively; R31 and R42 are water flows from the dam to male litter and from the dam to female litter, respectively. The model has five unknown parameters: i) K01 and K02 are equal and represent the output flow constants of the dam; ii) K03 and K04 are the output flow constants from male and female litter, respectively and iii) K31 and K42 are the output flow constants from the dam to its male and female litter, respectively.

Using the isotope dilution method, the mass of dam’s TBW (TBWd, in g) was calculated by dividing the amount of D_2_O injected by the D_2_O value extrapolated from the D_2_O concentration curve to the intercept with the y-axis at time 0 (Appendix A). TBWd (in %) was calculated by dividing the mass of TBWd (in g) by dams’ mean mass from L11 to L15. 

Flow constants of the model (K01, K03, K04, K31, and K42) were then determined using the SAAM II® software. As litter size (8 or 12 pups) presumably impacts D_2_O dilution in the litter, we used values of D_2_O mass instead of D_2_O concentrations for calculations in our model. To this end, D_2_O concentration, obtained after baseline concentration deduction, was multiplied by animal mass at each day of sampling, following Equation (1):(1)$$D_{2}O\;concentration\;\left(µg.g^{-1}\right)*animal\;mass\;\left(g\right)=D_{2}O\;mass\;\left(µg\right)$$

This calculation implicitly assumes that dams’ and pups’ TBW (in %) are equivalent, which is probably the case considering that suckled pups have about 75% of TBW [29], which is similar to suckling dams’ TBW [22]. Moreover, in order to indicate that compartment (1) and (2) represent the same individual (dam), the same data were incremented for both compartments, and K02 was forced to be equal to K01. Flow constants of the model were obtained directly with SAAM II® from a fit of plasma and urine D_2_O mass–time curves. The water flows, from mother to male litter R31 (g.h^−1^) and from the dam to female litter R42 (g.h^−1^), were calculated as the product of TBWd (g) and K31 or K42, respectively. These values were then multiplied by 24 to obtain the daily milk production (g.day^−1^).

In this model, R31 and R42 were associated to milk flows between the dam and its male litter or between the dam and its female litter, respectively, assuming milk was the only external source of water for the pups. Thus, milk flow corresponds to the milk produced by the dam, as determined from the D2O transfer from the dam to litter, after the dam received a D2O injection. As the water content of each individual milk was not accurately measured, milk flow was directly calculated from water flow without correction. We assumed that this does not alter conclusions concerning group comparisons. Total milk production of the dam was calculated by summing R31 and R42 and represented the mean milk production throughout the sampling period (L11 to L18). Milk consumption by male and female litters was obtained by dividing R31 and R42, respectively, by the number of males or females pups (4 in NP:8, LP:8 and LPF:8 groups or 6, in NP:12 and NPF:12 groups).

### 2.5. Milk Protein, Lactose, and Fatty Acid Analysis

Milk protein concentration was determined using a colourimetric Pierce BCA Protein Assay Kit (ThermoFisher Scientific, Waltham, MA, USA) with milk diluted at 1/40 in osmosed water and bovine serum albumin (fraction V) as standard. Milk lactose concentration was estimated using an enzymatic Lactose/D-Galactose Assay Kit K-LACGAR® (Megazyme, Bray, Ireland) with milk diluted at 1/20 in osmosed water and α-lactose monohydrate as standard. Milk total lipids were not fractionated. Fatty acids (FAs) were extracted using the modified liquid–liquid extraction method of Bligh-Dyer, as previously described [28]. Briefly, FAs were extracted from 30 μL milk in methanol–chloroform mix (1:1, *v/v*). Heptadecanoic acid was used as the internal standard. Total FAs were transesterified using boron trifluoride in methanol, and fatty acid methyl esters were analysed by gas chromatography using an Agilent Technologies 7890A® instrument. Each FA was expressed as a percentage of the total identified FAs, and only FAs whose percentage was above 0.5% were taken into account. The sum of FAs (in g.L^−1^) was assumed to represent total milk lipid (free fatty acids, triacylglycerols and phospholipids) content.

The energy content of milk was calculated by multiplying lactose, proteins, and lipid content by their energy content, assuming 4 kcal.g^−1^ for both carbohydrate (lactose) and protein and 9 kcal.g^−1^ for fat. Milk macronutrient production and milk energy production by the dam were then calculated by multiplying total milk production by macronutrient concentrations or energy content of milk respectively.

### 2.6. Dams and Offspring Metabolic Markers and Offspring Glucose Tolerance Test

At PND60, blood samples (300 µL) were collected from 6 h fasted offspring. An oral glucose tolerance test (OGTT) was performed in offspring at PND61 and PND62. Following 6 hours of fasting, all rats received a 2 g/kg BW dose of glucose by gavage. Blood samples (100 µL) were collected to determine plasma insulin concentration before (T0) and after 15 (T15) and 30 min (T30) gavage with glucose. Blood glucose was measured at T0, T15, T30, T45, T60, T90, and T120 after glucose intake, using a Performa Accu-Chek® glucometer (Roche Diabetes Care France, Meylan, France). 

Insulin (Rat Insulin ELISA kit®, ALPCO, Salem, USA), glucose (Glucose GOD FS®, DiaSys, Holzheim, Germany), triglycerides (Triglycerides FS®, DiaSys, Holzheim, Germany), and cholesterol (Cholesterol FS®, DiaSys, Holzheim, Germany) were measured in plasma, following manufacturers’ instructions. Optical density was read with a microplate reader Varioskan Lux® (ThermoFisher Scientific, Waltham, USA).

### 2.7. Trigonelline Quantification in Experimental Diet, Dams’ Plasma, and Milk

Ten µL of the labelled internal standard: trigonelline-D3 (CIL, Sainte Foy la Grande, France) at 50 µM, and 100 µL of acetonitrile were added to 10µL of diet solution (mixed diet in water, 100 g.L^−1^), dams’ plasma or milk. Samples were centrifuged for 10 min at 11,000 g, and the supernatant was injected into the LC–MS/MS using a hydrophilic interaction liquid chromatography system (Acquity H-Class® UPLC^TM^ device, Waters) coupled with a triple quadrupole mass spectrometry detector (Xevo® TQD) with an electrospray interface. Data acquisition and analyses were performed with MassLynx® and TargetLynx® software, both versions 4.1 (Waters). Trigonelline was separated over 6 min on a HILIC column (2.1 × 100 mm; 1.7 μM), (Waters) held at 45 °C with a linear gradient of mobile phase A (10 mM ammonium acetate in water) in mobile phase B (98% acetonitrile in water), each containing 0.1% formic acid, at a flow rate of 400 μL/min. Trigonelline was detected by the Xevo® TQD that allowed the multiple reaction monitoring (MRM) mode to be performed, with the electrospray interface operating in the positive ion mode (capillary voltage, 1.5 kV; desolvation gas (N_2_) flow and temperature, 650 L/h and 150 °C; source temperature, 150 °C). Based on specific collision-induced fragmentation of precursor ions, the MRM precursor/fragment pairs were based on the following transitions *m/z*: 138.06→93.98 amu for trigonelline and 141.03→97.00 amu for the internal standard. Chromatographic peak area ratios between trigonelline and its internal standard constituted the detector response. Trigonelline standard solutions (concentrations from 1 nM to 10 µM) were used for calibration.

### 2.8. Statistical Analysis

To maximise the power of analysis, parametric tests were favoured using one-way ANOVA or two-way ANOVA with an experimental group factor and a day or a sex factor. The validity of the parametric tests was checked by assessing normality of residuals with a Shapiro–Wilk test. In the case of absence of residual normality non-parametric, the Kruskal–Wallis test was used instead of one-way ANOVA. Multiple comparison tests used after ANOVA were: Dunnett’s post-hoc test to compare several levels to one level of interest (Table 1), Tukey’s post-hoc test to compare levels altogether (Table 2) and Sidak’s post-hoc test when there were only two levels in a factor (Table 4). After the Kruskal–Wallis test, the non-parametric Dunn’s post-hoc test was used. To determine the strength of the link between two variables, Pearson’s correlation tests were used, and simple linear regression was performed to determine the relation between two variables. All tests were performed with the GraphPad prism® software, version 6.

## 3. Results

### 3.1. Dams and Litter Characteristics in the Reference Group NP:8

For all groups, we took care to balance pups’ birth weight, following the adoption of NP pups and litter standardization. Mean pup birth weight was 7.1 ± 0.0 g, with no difference between groups (*p* = 0.37), although males were significantly bigger than females (7.3 ± 0.0 g versus 6.9 ± 0.0 g, *p* < 0.001). NP:8 dams weighed 344.3 ± 8.7 g at delivery. Maternal weight loss during the lactation period and food and water intakes are given in Table 1. Litter growth rate was 22.9 ± 0.7 g∙day^−1^, which represented a pup growth rate of 2.86 ± 0.09 g∙day^−1^ and did not differ between genders (*p* = 0.23). In the water turnover models, output flow constants (K3,1 and K4,2), as reported in Appendix A, were determined with good accuracy due to a forecast standard deviation (FSD) of each group lower than 5%, allowing the milk flows to be determined with confidence. For the NP:8 group, milk flow was 46.0 ± 1.6 g∙day^−1^, which represented a mean milk consumption of 5.74 ± 0.20 g∙day^−1^ per pup, with no significant sexual dimorphism (*p* = 0.36). 

### 3.2. Lactation Challenges Impact both Physiological Characteristics of Dam and Litter Compared to the Control Group NP:8

As no global sex effect was observed for pups’ growth and milk flow variables, results from the female and male offspring were pooled. Except for weight at delivery (similar for all groups, mean of 348.4 ± 5.2 g), every variable characterizing lactating dams, pups’ growth and milk flow was significantly affected by both lactation challenges (Table 1).

In the litter size increase challenge, dam’s weight loss during the entire lactation period was similar between NP:8 and NP:12 dams. Food and water intakes of lactating dams increased significantly (+17%) in NP:12, compared with the NP:8 group. Similarly, the litter growth rate between L11 and L18 was significantly increased (+21%). The increase of litter size led to extra-uterine growth restriction (EUGR) in the offspring during the lactation period since pup growth rate was 20% lower in NP:12, compared to the NP:8 litters. The enhancement in milk production with larger litter size remained insufficient to cope with increased demand: although total milk production of NP:12 dams was 18% greater than for NP:8 dams (*p* = 0.029), NP:12 pups’ milk consumption was 21% lower than in NP:8 pups (*p* < 0.001).

In the maternal protein restriction challenge, dam’s weight loss during the entire lactation period was 4.6-fold greater for LP:8 than for the NP:8 (*p* = 0.002) group. Food and water intakes of lactating dams were significantly decreased for LP:8 dams compared to NP:8 dams (−14% and −40%, respectively). Similarly, litter and pup growth rates were significantly decreased (−53%) in LP:8 pups, confirming EUGR occurred in response to maternal protein restriction. The maternal dietary protein restriction challenge was very effective since both values of total milk production, and milk consumption by pups was much lower for the LP:8 group than for the NP:8 group (−44%).

### 3.3. Correlation between Milk Flow Variables, Pups’ Growth Variables, and Lactating Dams’ Intakes Variables

Pearson’s correlations were calculated between milk flow variables and lactating dam’s intakes or pup’s growth variables. Total milk production was strongly correlated with litter growth rate (Figure 3a), dams’ food (*r* = 0.86, *p* < 0.001) and water (*r* = 0.93, *p* < 0.001) intakes. Similarly, pups milk consumption was strongly correlated with pup growth rate (Figure 3b).

### 3.4. Determination of Galactologue Effect of Fenugreek in Two Models of Lactation Challenges

Fenugreek intake was close to 1 g.kg BW^−1^.d^−1^ for both supplemented groups (i.e., 1.01 ± 0.02 g.kg BW^−1^.d^−1^ for NPF:12 dams and 0.93 ± 0.02 g.kg BW^−1^.day^−1^ for LPF:8 dams), but consumption was 7.8% higher in the NPF:12 group than in LPF:8 group (*p* = 0.018). Trigonelline was measured in the diet as a marker of fenugreek content, and was 12.5 µg.kg^−1^ and 14.9 µg.kg^−1^ for the NPF:12 and LPF:8 diet, respectively ( SD1 ), leading to a slightly higher trigonelline intake for NPF:12 (1.99 ± 0.05 mg.kg BW^−1^.day^−1^) than LPF:8 dams (1.73 ± 0.04 mg.kg BW^−1^.day^−1^).

In the litter size increase challenge, no significant difference was observed for NPF:12 dams on delivery weight (367.0 ± 11.1 g, *p* = 0.28), weight loss, or water intake during lactation when compared to NP:12 dams (Table 1), although food intake was 13.1% higher in NPF:12 than NP:12 dams *p* = 0.001). However, when intake was reported to dam’s body weight, the difference in food intake was no longer significant (153.8 ± 2.1 and 161.0 ± 3.6 g.kg^−1^.day^−1^, respectively). Moreover, water intake of NPF:12 dams (163.9 ± 3.7 g.kg^−1^.day^−1^) became significantly lower compared to NP:12 dams (175.8 ± 4.2 g.kg^−1^.d^−1^, *p* = 0.045), but was still significantly higher compared to NP:8 dams (146.8 ± 2.2 g.kg ^-1^.day^−1^, *p* = 0.008). 

In this challenge, fenugreek promoted pup growth when comparing NPF:12 and NP:12 groups (Figure 4a). This significant difference was observed from L12 (*p* = 0.022) to L18 (*p* = 0.002). However, NPF:12 pups failed to reach the growth observed in NP:8 pups. The final weight gain of NPF:12 pups was increased by 10.6% compared to NP:12 pups but was 12.2% lower compared to NP:8 pups (Figure 4b). 

Fenugreek promoted milk flow: total milk production was 16.1% higher in NPF:12 dams (63.0 ± 3.1 g.day^−1^), compared with NP:12 dams (*p* = 0.048) (Table 1). Milk consumption per pup also tended to be higher (*p* = 0.059) with a greater increase for males (+17.7%, *p* = 0.028) than females (+13.8%, *p* = 0.088), although no overall sex effect was observed (Figure 4c). 

In the maternal protein restriction challenge, fenugreek had no effect on dam’s weight loss during lactation, nor on water and food intakes when comparing with the non-supplemented group LP:8. No significant difference was also observed for pup growth between LPF:8 and LP:8 groups neither for growth rate (Table 1), nor for weight gain during overall lactation or at a specific day (Figure 4d,e). Fenugreek had no effect on milk production (Table 1) and pups milk consumption (Figure 4f).

### 3.5. Fenugreek Enhances Milk Lactose and Trigonelline Content in the Litter Size Increase Challenge

In the litter size increase challenge, milk composition was similar between the NP:8 and NP:12 groups. Fenugreek had no effect on milk fatty acids (FAs), protein, and energy content, but led to a 27% increase in lactose concentration compared to NP:12 (Table 2). Every macronutrient flow was significantly increased in the NPF:12 group compared to the NP:12 group (+29.6%, +49.1%, and +28.9% for protein, lactose and FAs, respectively), leading to a 30.5% increase in energy flow for the NPF:12 group, while no difference was observed between the NP:12 and NP:8 groups. 

The maternal protein restriction challenge had no significant effect on lactose and FAs concentrations and energy of milk but resulted in a significant 16% decrease in protein concentration (LP:8 vs. NP:8). Flow was significantly decreased for every macronutrient and for energy in the LP:8 group compared to NP:8 (−52.8%, −51.1%, −34.0%, and −38.9% for protein, lactose, FAs, and energy, respectively). Fenugreek failed to induce a modification in macronutrient composition or energy (in concentration and in flow) between LPF:8 and LP:8 milk.

Trigonelline concentration was measured in milk at L18 as a marker of fenugreek content. NPF:12 milk had a trigonelline concentration 17.1-fold higher than NP:12 milk (2575.6 ± 132.9 nM and 150.3 ± 22.9 nM, respectively, *p* < 0.001). Similarly, LPF:8 milk had a significant 5-fold increased trigonelline concentration compared to LP:8 milk (2683.2 ± 141.9 nM and 533.7 ± 118.2 nM respectively, *p* < 0.001). 

### 3.6. Effect of Fenugreek on Metabolic Status of Dams and Offspring at Short and Long Term for the Litter Size Increase Challenge

As fenugreek had a positive effect on milk production and quality only in the litter size increase lactation challenge, the metabolic status of dams and offspring was assessed exclusively in the NP:8, NP:12, and NPF:12 groups.

Dams’ metabolic parameters were determined in plasma sampled at L12 without a fasting period and at L21 after a 4 h-fasting period (Table 3). Milk cholesterol, triglycerides (TGs), and glucose concentrations rose significantly between L12 (mid lactation) and L21 (end of lactation). A larger rise in cholesterol was observed in the NPF:12 group (+20,7%, *p* < 0.001) compared to the NP:12 group (+5%), but no significant difference was observed between the NPF:12 and NP:12 groups, at both days of lactation. Insulin concentration was significantly lower in the NPF:12 group than in NP:12 group (−42.4%, *p* = 0,018) transitory at L12, but no significant difference was observed between the NPF:12 group and NP:8 group.

As expected, trigonelline content was largely increased at L12 in NPF:12 dams’ plasma compared to NP:12 (3258.7 ± 200.4 nM and 464.3 ± 78.5 nM respectively, *p* < 0.001).

Offspring’s metabolism was determined in the short (PND20) and long-term (PND60) when pups reached early adulthood. Selected plasma parameters of lipid and glucose metabolism are presented in Table 4. At PND 20, lactation challenge by litter size increase did not affect cholesterol, TGs, or insulin concentration in offspring’s plasma, whereas it tended to increase glucose concentration (+7.5%, *p* = 0.052) in the NP:12 group vs. the NP:8 group. At PND 60, it tended to a decrease TGs concentration (−17.8%, *p* = 0.10) mainly for males (−21.7%, *p* = 0.047). Fenugreek had no effect on glucose, insulin, and TGs concentration although, at PND 20, a significant decrease in plasma cholesterol concentration was observed in the NPF:12 group compared to the NP:12 group (−10.0%, *p* = 0.024), greater for female (−11.8%) than for male (−8.1%). Fenugreek no longer had any effect on cholesterol concentration at PND 60.

Long-term glucose metabolism was assessed in offspring by OGTT only for NP:12 and NPF:12 groups (Figure 5). Before oral glucose gavage (T0), no significant difference was observed between NPF:12 and NP:12 groups for plasma glucose and insulin concentrations, but females had significantly lower concentrations than males (−25.0%, *p* = 0.014), greater in the NPF:12 group (−37.1%, *p* = 0.064) than in the NP:12 group (−17.4%, *p* = 0.33). After oral glucose gavage, no difference was observed in glycaemia neither between sexes nor between groups (Figure 5a,b). Fenugreek decreased the insulin area under curve (AUC) in NPF:12 compared to the NP:12 group (−38.4%, *p* = 0.004) (Figure 5d) and this difference was greater for females (−60%, *p* = 0.044) than for males (−29.7%, *p* = 0.094). Insulin peak was delayed in the NPF:12 (30 min) group compared to the NP:12 group (15 min) (Figure 5c).

## 4. Discussion

To the best of our knowledge, this study is the first to explore the effect of fenugreek on milk production in a rat model. The effect on milk production, measured using stable isotope labelled water, was assessed in two separate models of lactation challenge: (a) litter size increase from 8 to 12 pups, and (b) maternal, perinatal dietary protein restriction (from 20% to 8%). We found that when dams were under appropriate physiological conditions for lactation and confronted with a litter size increase, fenugreek produced an increase in milk flow. In contrast, when dams were placed under inappropriate physiological conditions of lactation following dietary protein restriction, fenugreek was ineffective.

### 4.1. Effect of Lactation Challenges on Pup Growth and Milk Production

First, we verified that litter size increase and maternal, perinatal protein restriction both led to a decrease in milk consumption by the pups, resulting in Extra-Uterine Growth restriction (EUGR), confirming that dams had difficulties producing milk sufficiently to meet the demand. 

When litter size was increased from 8 to 12 pups, dams adjusted their milk production, suggesting dams’ adaptations to the increased demand from pups. Indeed, the litter size increase led to a 21% increase in litter growth rate, along with an 18% increase in milk flow. These results are in accordance with those from Morag et al. [23] and Kumaresan et al. [30] who reported a 43% and 22% increase of milk yield, measured by weight-suckle weight method, for a litter size increase from 9 or 8 pups to 12 pups, respectively. However, this rise of milk production was not sufficient to compensate the litter size increase, resulting in EUGR with a 26% lower overall pup weight gain, presumably due to a 21% decrease in milk consumption per pup. Kumaresan et al. [30] found similar results, with a 20% decrease in milk availability per pup at L14 when litter size was increased from 8 to 12 pups, and an 11% decreased in pup weight at L18 (31.1 g for 8 pups to 27.6 g for 12 pups) compared to the 18% decrease in our study (52.3 g for NP:8 and 42.9 g for NP:12 at L18). Our values of pup growth and milk consumption are probably closer to physiological values than those of Morag et al. [23] and Kumaresan et al. [30] as we considerably reduced the stress of dam/pup separation (from 8 h to 10 h in these studies compared to 30 min in ours). These conditions in which the dams’ capacity to increase milk production is preserved but insufficiently to produce optimal pup growth, as observed in a litter of 12 or more pups, are likely the most suitable to test the effect of a galactologue compound [23,30].

In the other lactation challenge (maternal dietary protein restriction), we confirmed that 8% perinatal protein restriction led to an EUGR due to an impaired ability of dams to produce milk. This challenge led to a 44% reduced milk flow, confirming data from our earlier studies [22], and resulted in impaired pup growth with a 42% decrease in pup weight gain at L18 (as already observed by Bautista et al. [25] and Martin-Agnoux et al. [31]). The impaired ability of mothers to produce milk suggests the dams’ physiological status was impaired due to undernutrition during gestation and lactation. Indeed, during lactation, LP:8 dams exhibited a 4.5-fold greater weight loss than NP:8 dams, which is a well-described consequence of nitrogen store depletion on milk-protein synthesis in states of malnutrition [32,33]. We also observed a significant 36% reduction in mammary gland weight at the end of lactation in the LP:8 group compared to the NP:8 group (5.73 ± 0.67 g for NP:8 vs. 3.67 ± 0.75g for LP:8 dams), as already reported in protein-restricted dams [32]. Finally, secretion of prolactin, the principal hormone promoting milk synthesis, has been shown to decrease by 70% in the serum of dams fed a low protein diet [32]. Altogether, these physiological changes contribute to a decrease in dams’ capacity to produce sufficient milk to meet the offspring requirement.

### 4.2. Correlation between Milk Flow, Pups’ Growth, and Lactating Dams’ Intakes of Food and Water

Secondly, we confirmed that the use of the water turnover method, with D_2_O values as mass, reliably measures milk flow. When considering the three lactation models altogether, the changes in total milk production were strongly correlated with litter growth rate (*r* = 0.93), and changes in milk production and milk consumption accounted for at 87% and 82% of the change in growth rate in the two models tested (Figure 3). As litter or pup growth rate and milk production or consumption are expressed in the same unit (g.day^−1^), the slopes of 0.51 with intercept close to 0 in both regression lines suggest that the consumption of 1g of milk produced a weight gain of 0.51 g per day between L11 and L18. These results are also supported by the strong correlations between milk production and the dams food and water intake (*r* = 0.86 and *r* = 0.91, respectively), representing the increasing needs of the lactating mother for milk production with large litters [33]. Altogether these results suggest that the water turnover method, with values of D_2_O mass, accurately measures milk production regardless of the dam conditions.

### 4.3. Galactologue Effect of Fenugreek in Two Models of Lactation Challenge

In the litter size increase challenge, dietary supplementation of fenugreek at a dose of 1 g.kg^−1^.d^−1^ increased milk flow by 16% and increased pup growth rate and final weight gain, by 9% and 11%, respectively. The galactologue effect of fenugreek is consistent with the rise in milk production observed in other mammals (+13% at a fenugreek dose of 2 g.kg^−1^.day^−1^in goat [18] and +18% at a fenugreek dose of 270 mg.kg^−1^.d^−1^ in buffalo [17]). Other authors found a stronger effect of fenugreek with a 42% greater pup growth at the end of lactation in rabbit (fenugreek dose of 0.5 g.kg^−1^.day^−1^) [16] and a 110% increase in milk yield in ewe (fenugreek dose of 1.2 g.kg^−1^.day^−1^) [19]. The stronger galactologue effects in the rabbit could be explained by the fact that they only nurse their litter once a day [24], suggesting that pups have important growth with only a small milk intake. Otherwise, in ewes, fenugreek supplementation is associated with greater crud protein and energy content in the ration compared to the control group, which could also affect milk production. 

In contrast, in the maternal protein restriction challenge, fenugreek failed to affect pup growth or milk production. Undernutrition likely produced profound alterations in global metabolism and mammary gland development and function that could not be reversed by a galactologue.

The mechanisms underlying the impact of fenugreek in the litter size challenge raise many questions. The reported increase in plasma prolactin and growth hormone may play an important role [17,18]. Fenugreek might also act by allowing dams to maintain their weight during lactation. Indeed, a significant correlation was found between dam mass and pup growth in rats [24]. Yet, NPF:12 dams’ weight at L11 was significantly higher than NP:12 dams (*p* = 0.021) although no significant difference was observed at L0. Increased food intake in the fenugreek-supplemented group likely played a minor role in maintaining dam weight since the increased energy intake barely compensated for the increasing energy output that accounted for increased milk production. Thus, fenugreek probably promotes energy storage, which in turn could positively affect the mother’s lactation performance [24]. Finally, the effect of fenugreek may be mediated by trigonelline [15]. Indeed, a 7-fold rise in dam plasma trigonelline was observed following fenugreek supplementation at 1 g.kg^−1^.day^−1^. Trigonelline is a precursor of niacin or nicotinic acid (B3 vitamin) involved in the formation of nicotinamide adenine dinucleotide (NAD^+^) [34]. This coenzyme factor may play a key role in various metabolic pathways such as i) ATP formation, via its reduction in NADH in glycolysis, beta-oxidation, and citric acid cycle [35], ii) cell survival, as NAD^+^ is the sole precursor of PARP, a DNA repair enzyme, and iii) transcriptional regulation as it is a main cofactor of sirtuins [36]. Thus, in the mammary gland, an increase in trigonelline could increase NAD^+^ content, and thus, enhance lactation by promoting energy supply for milk synthesis, as well as mammary cell longevity and function.

### 4.4. Effect of Fenugreek on Milk Composition

A separate collection of milk secreted at the beginning and at the end of each suckling is not feasible. Milk composition was assessed at L18, which corresponds to the end of the lactation period in the rat. As rat milk composition changes during lactation [37], macronutrient composition reported at L18 may not reflect day to day variations in milk composition during the entire period during which the milk production was determined (L11–L18). As observed by earlier studies [38], the increase in litter size did not alter milk macronutrient concentration. Fenugreek increased milk lactose by 27%, whereas lipids and proteins remained unchanged. Discrepant results of fenugreek supplementation on milk macronutrient content have been reported in the literature in several animal models (rabbit, buffalo, goat, and ewe) [16,17,18,19]. However, lactose was the most affected by fenugreek, especially in our model, and its concentration is increased in most cases [16,17]. The key osmotic regulatory role of lactose on milk secretion could explain the positive effect of fenugreek on total milk production through an increase in water flow from the mammary epithelial cells into the mammary secretory vesicles and subsequently into the alveolar lumen [39]. 

Increasing litter size did not increase macronutrient and energy flow although milk flow was significantly increased, suggesting that the increase in milk production was offset by a dilution of macronutrients, likely due to the osmotic role of lactose. Indeed, NP:12 milk had the lower mean value of protein, fat, and energy, leading to even lower intakes for the pups. In contrast, macronutrient and energy flows were all increased when dams were supplemented with fenugreek suggesting that, with the increase in total milk production, the activity of the synthesis pathways of the three milk macronutrients was enhanced to maintain baseline concentrations. Further molecular investigations would be needed at the mammary gland level to confirm this hypothesis. In terms of pups’ intake, the increase in macronutrient and energy flow led to a similar intake of the three macronutrients by the NPF:12 pups and NP:8 pups (Appendix A). As lactose concentration was increased in NPF:12 milk, the mean lactose intake by NPF:12 pups was slightly, but not significantly larger than the lactose intake in NP:8 pups. Larger lactose intake, however, did not allow NPF:12 pups to achieve the same growth rate as NP:8 pups. This suggests that lactose is not the major nutrient that promotes pup growth. Moreover, as the three macronutrients were consumed in similar amounts in both groups despite different growth trajectories, this suggests that micronutrients and/or other bioactive compounds of milk likely impact pup growth. In the maternal protein restriction model, fenugreek had no effect on milk composition. Once again, important modifications of milk composition due to maternal undernutrition [28] cannot be overcome by fenugreek.

### 4.5. Effect of Fenugreek on Dams Metabolism during Lactation and on Short- and Long-Term Offspring Metabolism

At mid-lactation, fenugreek did not modify dams’ plasma cholesterol, triglycerides, and glucose concentrations as previously found in ewe [19] or in goat [21], suggesting that fenugreek did not alter maternal blood metabolic biomarkers during lactation. However, plasma insulin concentration was significantly increased in the NP:12 group and returned near to the baseline level in the NPF:12 group. Among metabolic adaptations reported during lactation, mammary gland displays enhanced insulin sensitivity while other tissues developed insulin resistance, leading to a redirection of energy substrates toward the mammary gland [40]. These changes result in a decrease in plasma insulin concentration, which is inversely related to milk production [41] and is consistent with the increased milk production and lower insulin concentration observed in NPF:12 dams at L12.

Contrary to the overall lactation period, the suckling phase leads to a rise in plasma insulin concentration [41,42]. Yet, plasma was sampled just after dam/pup separation, and pups were allowed to suckle until separation. The greater number of pups in NP:12 litters compared to NP:8 litters probably led to greater suckling, explaining the greater dams’ insulin level observed. All these results in insulin must be relativized, knowing that at L12, dams’ plasma was sampled without a fasting period.

At the end of the lactation period, both cholesterol and triglycerides were greatly increased in NPF:12 than in NP:12 dams’ plasma compared to mid-lactation, which could be explained by the higher milk production of NPF:12 dams. Indeed, during lactation, the mammary gland becomes the main site of lipogenesis with a rate 5-fold higher than in liver [33], which in contrast, increases its hepatic cholesterol synthetic activity [43] and in detriment of adipose tissue, which exhibits an increased lipolytic activity [33]. These changes are associated with a large rise in triglyceride and cholesterol uptake from circulating chylomicrons in the mammary gland thanks to the increased activity of prolactin-mediated lipoprotein lipase (LPL) [44]. At the end of lactation, the prolactin-mediated LPL activity in the mammary gland drops rapidly [43], although lipolytic activity in adipose tissues and cholesterol synthesis in liver persist for a while. This leads to higher plasma triglyceride [43] and cholesterol concentration, which could presumably be further increased when milk production is enhanced. This would be consistent with the differences observed in NP:12 and NPF:12 plasma and with the larger rise in plasma triglyceride concentration in NP:12 dams compared to NP:8 dams. 

Finally, the current study was the first to demonstrate the absence of adverse effects of dams’ fenugreek supplementation on offspring metabolism. In the short-term, only cholesterol plasma concentration was decreased by fenugreek supplementation. Fenugreek is known to have a hypocholesterolemic effect that has been attributed to the large amount of fibre in fenugreek seed [45], or to steroid saponin such as diosgenin [15]. The effect of fenugreek on plasma cholesterol concentration of the offspring in the short-term may be mediated by the consumption of these components in milk. Indeed, milk trigonelline concentration increased 17-fold upon maternal supplementation, implying many components of fenugreek appear in mothers’ milk. Alternatively, as pups begin to eat dams’ pellet food at the end of lactation, the hypocholesterolemic effect of fenugreek might be directly mediated by solid food consumption. When fenugreek consumption stopped at L20, cholesterol concentration became indistinguishable from those of control offspring at PND 60, suggesting only a short-term hypocholesterolemic effect of fenugreek.

In the long run, no difference was observed between NPF:12 and NP:12 offspring in glucose concentration after glucose gavage, although NPF:12 offspring had lower concentrations of insulin with a delayed peak. Although a delayed peak of insulin is generally associated with insulin resistance [46], it is often related to an increased glucose concentration and area-under-curve, which is not the case in our study. Conversely, the decrease in insulin area under curve could reflect greater insulin sensitivity, and fenugreek is known to have antidiabetic properties [15], notably via the action of trigonelline [47]. The increased consumption of milk trigonelline by pups during lactation may help them to develop higher insulin sensitivity in the short-term and thus favour long-term glucose metabolism. However, despite the apparent absence of metabolic alterations, fenugreek should not be consumed by women with asthma, digestive disorders, hypertension, heart cardiovascular disease, or hypothyroidism because of its possible side effects [15,48].

## 5. Conclusions

The current study confirms the galactologue effect of fenugreek in another mammal model, suggesting that this dietary supplement may be helpful in humans. We tested the capacity of dams’ fenugreek to increase milk production and subsequent pup growth in two models of the lactation challenges. The galactologue effect of fenugreek was confirmed, but only in the challenge by litter size increase when dams had no physiological impairment in their ability to produce milk. In contrast, the lack of effect of fenugreek under maternal dietary protein restriction suggests that fenugreek cannot overcome the lactation impairment due to undernutrition. Thus, fenugreek supplementation might enhance milk production in the case of insufficient maternal milk production, due to maternal stress, difficulties in breastfeeding management, first parity, or when mothers are breastfeeding twins, but fenugreek is unlikely to be effective in situations that affect lactation physiology, such as undernutrition deficiency, mammary hypoplasia, and hormonal deregulation. Finally, we observed no adverse metabolic effect neither on the dam at mid- and end-lactation nor on offspring, with preliminary evidence that fenugreek might even enhance insulin sensitivity in the long run. The 16% increase in milk production, thanks to fenugreek, is of significance and clearly warrants the design of clinical trials in breastfeeding women

## Figures and Tables

**Figure 1 nutrients-11-02571-f001:**
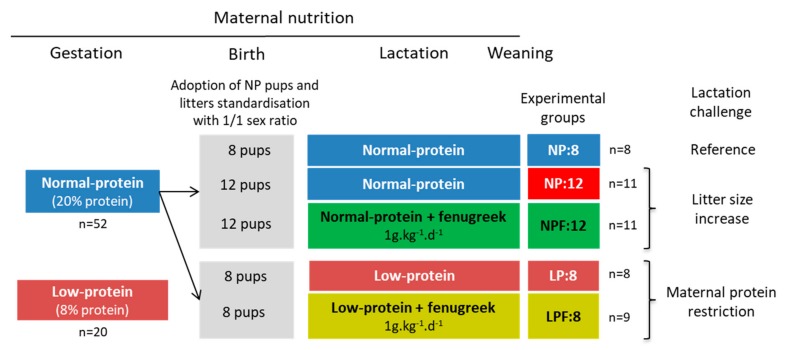
Representation of the 5 experimental groups of dams.

**Figure 2 nutrients-11-02571-f002:**
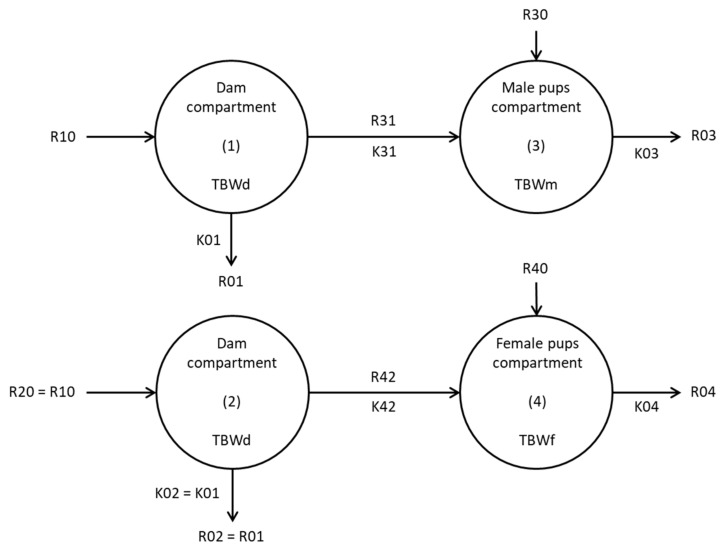
Four-compartment model of water turnover. TBWd, TBWm, TBWf: total body water (expressed in g) of dam, male, and female litter respectively; K01 and K02 (in h^−1^) are equal and represent the output flow constants from the dam; K03 and k04 (in h^−1^) are the output flow constants from the male and female litters, respectively; K31 and K42: (in h^−1^) are the output flow constants from the dam to its male litter and from the dam to its female litter, respectively; R10, R20, R30, and R40 (in g.h^−1^) are input water flows into the body of dam (R10 is equal to R20) and its male and female litter respectively; R01 and R02 (in g.h^−1^) are equal and represent output water flows from the dam; R03 and R04 (in g.h^−1^) are output water flows from male and female litters respectively; R31 and R42 (in g.h^−1^) are water flows from the dam to its male litter and from the dam to its female litter and are associated to milk flow.

**Figure 3 nutrients-11-02571-f003:**
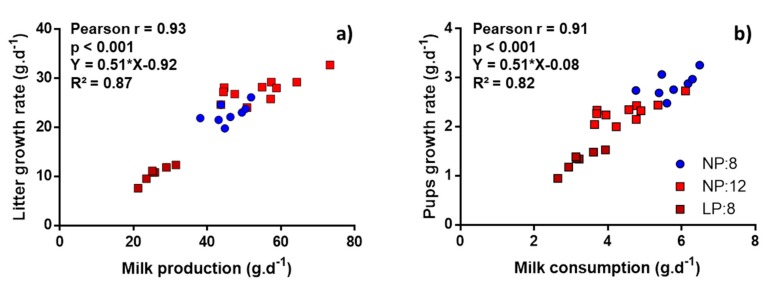
Correlation between pup growth and milk flow variables in three different conditions of lactation physiology (NP:8, NP:12, and LP:8). (**a**) Correlation between total milk production and litter growth rate from L11 to L18. (**b**) Correlation between pup milk consumption and pup growth rate from L11 toL18. NP:8, 20% protein diet with 8 pups per litter; NP:12, 20% protein diet with 12 pups per litter, and LP:8 groups, 8% protein diet with 8 pups per litter.

**Figure 4 nutrients-11-02571-f004:**
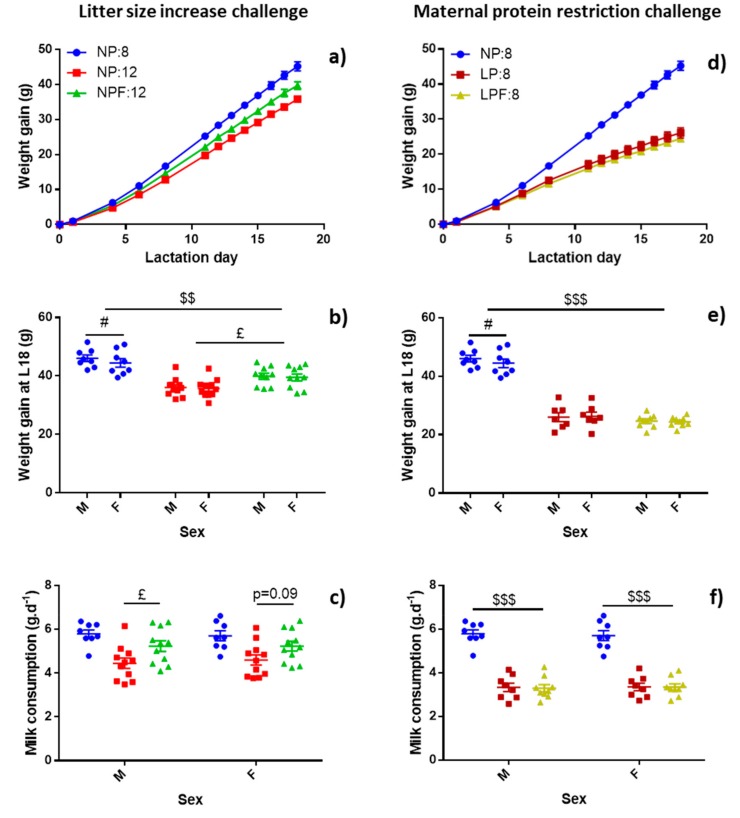
Effect of fenugreek on pup growth and pup milk consumption in 2 models of lactation challenges by litter size increase (**a**,**b**,**c**) or by maternal protein restriction (**d**,**e**,**f**). Weight gain during lactation was represented in graphs (**a**) and (**d**). Values were mean of male and female pups’ weight gain. Final weight gain at L18 was represented in graphs (**b**) and (**e**) and pup milk consumption was represented in graphs (**c**) and (**f**) for both males (M) and females (F). Results were analysed with two-way ANOVAs with group and day factors for graphs (**a**) and (**d**) and with group and sex factors for other graphs. Pairwise comparisons were realised with Dunnett’s post-hoc test to compare fenugreek supplemented groups to their challenge control and NP:8 and with Sidak’s *post-hoc* tests for sex factors. £ *p* < 0.05 represented the significant difference between NPF:12 or LPF:8 and their own lactation challenge model control. $$ *p* < 0.01 and $$$ *p* < 0.001 represented the significant difference with NPF:12 or LPF:8 and NP:8. # *p* < 0.05 represented the significant difference between male and female pups.

**Figure 5 nutrients-11-02571-f005:**
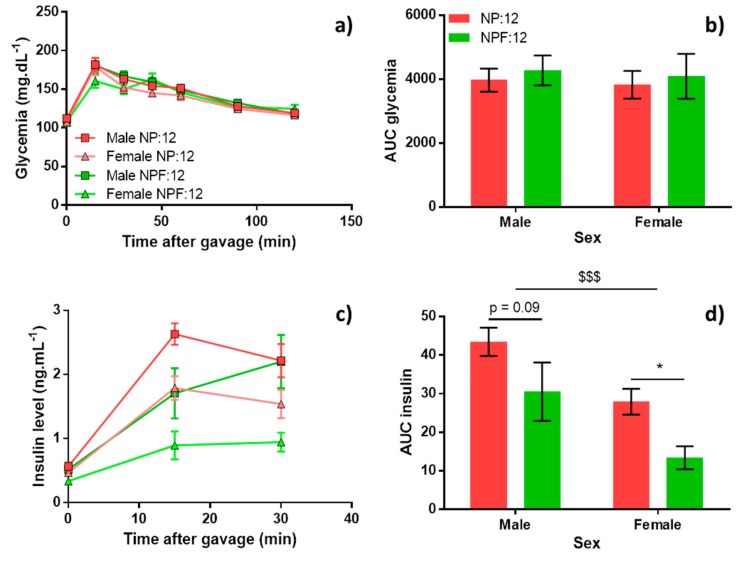
Effect of fenugreek on long-term glucose metabolism assessed by oral glucose tolerance test for NP:12 and NPF:12 groups. At PND 60, after 6 h fasting, 2 g.kg^−1^ of glucose was injected to rats by gavage. (**a**) Time course of glycaemia after glucose gavage; (**b**) area under curve (AUC) of glycaemia; (**c**) time course of insulin concentration after glucose gavage; (**d**) AUC of insulin concentration. For NP:12, *n* = 12 per sex and for NPF:12, *n* = 8 per sex. Values were mean ± SEM and were analysed with two-way ANOVA with group and sex factors followed by Sidak’s post-hoc test. $$$ *p* < 0.001 represents significant difference between sexes and * *p* < 0.05 represents significant difference between groups.

**Table 1 nutrients-11-02571-t001:** Effect of fenugreek on dams’ lactation follow up, pups growth, and milk flow in two models of lactation challenges. Results are expressed as mean ± SEM.

Lactation Challenges	*Control*	*Litter Size Increase*		*Maternal Protein Restriction*	
Experimental Groups	NP:8	NP:12	NPF:12	LP:8	LPF:8
*n*	8	11	10	7	9
**Dams** (L0–L21)					
Weight loss, g	–6.2 ± 2.4	–13.5 ± 4.0	–7.7 ± 2.9	–8.0 ± 3.7 **	–7.9 ± 2.2 ^$$$^
Food intake, g.d^−1^	43.3 ± 1.5	50.7 ± 1.0 ***	57.4 ± 1.2 ^$$$ £££^	37.2 ± 1.3 **	37.7 ± 0.7 ^$$^
Water intake, g.d^−1^	49.6 ± 1.5	58.0 ± 1.4 ***	58.5 ± 2.0 ^$$^	30.3 ± 1.5 ***	29.1 ± 1.1 ^$$$^
**Pup growth** (L11–L18)					
Litter growth rate, g.d^−1^	22.9 ± 0.7	27.6 ± 0.7 ***	30.1 ± 0.8 ^$$$ £^	10.3 ± 0.6 ***	10.0 ± 0.3 ^$$$^
Pups growth rate, g.d^−1^	2.86 ± 0.09	2.30 ± 0.06 ***	2.51 ± 0.06 ^$$ ▪^	1.31 ± 0.08 ***	1.25 ± 0.04 ^$$$^
**Milk flow** (L11–L18)					
Total milk production, g.d^−1^	46.0 ± 1.6	54.3 ± 2.8 *	63.0 ± 3.1 ^$$$ £^	25.9 ± 1.3 ***	26.6 ± 1.2 ^$$$^
Pups milk consumption, g.d^−1^	5.74 ± 0.20	4.52 ± 0.23 ***	5.25 ± 0.25 ^▪^	3.24 ± 0.16 ***	3.32 ± 0.16 ^$$$^

NP:8, 20% protein diet with 8 pups per litter; NP:12, 20% protein diet with 12 pups per litter, NPF:12, 20% protein diet with fenugreek (1 g.kg BW^−1^.d^−1^) with 12 pups per litter, LP:8, 8% protein diet with 8 pups per litter and LPF:8, 8% protein diet with fenugreek (1g.kg BW^−1^∙day^−1^) with 8 pups per litter. Dams and pup growth results for each individual are mean value through the indicated period (L0 to L21 or L11 to L18). The two lactation challenges (NP:12 and LP:8) were compared to NP:8 with one-way ANOVAs followed by Dunnett’s post-hoc tests. * *p*<0.05, ** *p*<0.01 and *** *p*<0.001 compared to NP:8 group. Each fenugreek group (NPF:12 and LPF:8) were compared to its lactation challenge control (NP:12 or LP:8) and to NP:8 with one-way ANOVA followed by Dunnett’s post-hoc test. $ *p* < 0.05, $$ *p* < 0.01, $$$ *p* < 0.001 compared to NP:8 and ^▪^
*p* < 0.10, £ *p* < 0.05, ££ *p* < 0.01, £££ *p* < 0.001 compared to challenge model control.

**Table 2 nutrients-11-02571-t002:** Effect of fenugreek on energy and macronutrient composition of milk and on energy and macronutrient flows.

Lactation Challenges	Control	Litter Size Increase		Maternal Protein Restriction	
Experimental Groups	NP:8	NP:12	NPF:12	LP:8	LPF:8
*n*	7	8	9	7	9
**Macronutrient concentration,** g.L^−1^					
Protein	97.1 ± 4.9	91.7 ± 1.5	98.3 ± 3.3	81.5 ± 2.2 **	77.4 ± 2.1 ^$$$^
Lactose	30.1 ± 1.9	29.4 ± 1.2	37.4 ± 0.9 ^$$ £££^	27.4 ± 1.3	25.3 ± 1.1
Fatty acids	147.1 ± 12.8	129.4 ± 6.6	137.8 ± 7.2	174.3 ± 13.6	176.5 ± 9.2
**Energy**, kcal.dL^−1^	183.2 ± 12.0	165.1 ± 5.5	178.3 ± 6.9	200.4 ± 11.7	199.9 ± 8.8
**Macronutrient** **flow**, g.day^−1^					
Protein	4.49 ± 0.32	4.87 ± 0.26	6.31 ± 0.41 ^$$ £^	2.12 ± 0.12 ***	2.06 ± 0.12 ^$$$^
Lactose	1.38 ± 0.09	1.60 ± 0.12	2.38 ± 0.10 ^$$$ £££^	0.71 ± 0.04 ***	0.68 ± 0.06 ^$$$^
Fatty acids	6.79 ± 0.66	6.81 ± 0.31	8.78 ± 0.53 ^$ £^	4.48 ± 0.34 *	4.72 ± 0.42 ^$^
**Energy flow**, kcal.day^−1^	84.6 ± 6.6	87.2 ± 4.0	113.8 ± 6.2 ^$$ ££^	51.6 ± 3.0 ***	± 4.4 ^$$$^

Values were mean ± SEM and were analysed by one-way ANOVA followed by a Tukey post-hoc test to compare all groups of each lactation challenge. * *p* < 0.05, ** *p* < 0.01, *** *p* < 0.001 represented significant differences between NP:12 or LP:8 and NP:8. £ *p* < 0.05, ££ *p* < 0.01, £££ *p* < 0.001 represented significant differences between NPF:12 or LPF:8 and their own lactation challenge control.$ *p* < 0.05, $$ *p* < 0.01, $$$ *p* < 0.001 represented significant differences between NPF:12 or LPF:8 and NP:8.

**Table 3 nutrients-11-02571-t003:** Effect of fenugreek on metabolic parameters in dams’ plasma at L12 and L21 in the litter size increase lactation challenges.

Parameter	Groups			Two-Way ANOVA			
	NP:8	NP:12	NPF:12	Global Effects			
*n*	8	11	11	Inter	Group	Day	
Cholesterol, mg.dL^−1^				
L12	101.3 ± 6.0 ^a,1^	94.6 ± 3.9 ^a,1^	92.2 ± 3.8 ^a,1^	0.050		0.57	0.002
L21	105.8 ± 5.6 ^a,1^	99.2 ± 4.1 ^a,1^	111.3 ± 6.9 ^a,2^				
Triglycerides, mg.dL^−1^						
L12	54.6 ± 7.0 ^a,1^	43.5 ± 2.8 ^a,1^	52.8 ± 5.2 ^a,1^	0.027		0.062	<0.001
L21	132.0 ± 16.6 ^a,2^	189.9 ± 25.1 ^ab,2^	224.3 ± 23.7 ^b,2^				
Glucose, mg.dL^−1^						
L12	125.1 ± 6.4 ^a,1^	120.6 ± 3.7 ^a,1^	117.4 ± 3.7 ^a,1^	0.72		0.62	<0.001
L21	148.0 ± 3.1 ^a,2^	150.7 ± 6.5 ^a,2^	144.7 ± 6.2 ^a,2^				
Insulin, ng.mL^−1^						
L12	0.60 ± 0.09 ^a,1^	1.65 ± 0.25 ^b,1^	0.95 ± 0.26 ^a,1^	0.12		0.004	0.22
L21	0.68 ± 0.06 ^a,1^	0.98 ± 0.09 ^a,2^	0.94 ± 0.15 ^a,1^				

Values were mean ± SE and were analysed with two-way ANOVA with group and day factors and with repeated values for day factor. ANOVA was followed by Tuckey’s post-hoc test for comparisons between groups and by Sidak’s post-hoc test for comparisons between days. For each biomarker, different letters represented significant differences (*p* < 0.05) between groups at each day and the difference between numbers represented significant difference (*p* < 0.05) between days for each group.

**Table 4 nutrients-11-02571-t004:** Effect of fenugreek on offspring’s metabolism in the short and long term.

Parameter	Groups			Two-way ANOVA		
	NP:8	NP:12	NPF:12	Global Effects		
	PND 20			Inter	Group	Sex
*n* for each sex	8	22	11			
Cholesterol, mg.dL^−1^						
Male	143.2 ± 10.1 ^a,1^	144.5 ± 3.9 ^a,1^	132.8 ± 3.0 ^a,1^	0.225	0.005	0.039
Female	167.5 ± 12.4 ^a,1^	151.5 ± 4.1 ^a,b,1^	133.5 ± 6.1 ^b,1^			
Triglycerides, mg.dL^−1^						
Male	260.2 ± 39.2 ^a,1^	272.0 ± 21.1 ^a,1^	258.8 ± 56.3 ^a,1^	0.853	0.763	0.969
Female	284.5 ± 46.1 ^a,1^	267.8 ± 23.3 ^a,1^	235.0 ± 35.9 ^a,1^			
Glucose, mg.dL^−1^						
Male	157.1 ± 7.4 ^a,1^	164.4 ± 3.8 ^a,1^	164.5 ± 3.5 ^a,1^	0.446	0.055	0.655
Female	152.7 ± 3.7 ^a,1^	168 ± 3.0 ^a,1^	159.7 ± 3.5 ^a,1^			
Insulin, ng.mL^−1^						
Male	0.19 ± 0.04 ^a,1^	0.35 ± 0.05 ^a,1^	0.41 ± 0.09 ^a,1^	0.958	0.037	0.282
Female	0.28 ± 0.04 ^a,1^	0.41 ± 0.05 ^a,1^	0.46 ± 0.07 ^a,1^			
	**PND 60**					
*n* for each sex	10	22	16			
Cholesterol, mg.dL^−1^						
Male	76.8 ± 2.5 ^a,1^	85.0 ± 2.5 ^a,1^	81.5 ± 2.1 ^a,1^	0.433	0.208	0.170
Female	84.8 ± 3.1 ^a,1^	85.7 ± 2.6 ^a,1^	82.3 ± 3.0 ^a,1^			
Triglycerides, mg.dL^−1^						
Male	135.8 ± 11.7 ^a,1^	106.4 ± 7.0 ^b,1^	123.2 ± 12.0 ^ab,1^	0.390	0.095	<0.001
Female	66.0 ± 3.7 ^a,2^	59.5 ± 5.2 ^a,2^	64.1 ± 6.3 ^a,2^			

Values were mean ± SE and were analysed with two-way ANOVA with group and sex factors followed by Tukey’s post-hoc test for comparisons between groups and by Sidak’s *post-hoc* test for comparisons between sexes. For each biomarker, different letters represented significant (*p* < 0.05) differences between groups for each sex and different numbers represented significant differences (*p* < 0.05) between sexes for each group.

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
