# Peer review of "Impact of Fenugreek on Milk Production in Rodent Models of Lactation Challenge"

_nutrients, 2019, doi:10.3390/nu11112571_

Round 1
Reviewer 1 Report
A well-thought out and very thorough study-- congratulations to the authors! You've tackled some important questions-- what is the efficacy of fenugreek under normal versus challenging circumstances? And what impact, if any, does it have on the offspring? This is sorely needed information for human mothers, and a rat study is a great start to providing this. Thank you for using a dosage corresponding to that most commonly used for breastfeeding purposes.
I have made some comments below, the rest are in the text itself.
English grammar- some word choices, and definitely some poorly placed punctuation, caused me to pause and work out the meaning a number of times. I have noted them in several places and in many cases made suggestions for better wording.
Re: fenugreek supplementation. Is this a supplement, or is it a therapeutic treatment? The term "supplement" is not typically used for human herbal galactogogue studies. In many cases throughout the text you can easily drop "supplementation" and just say "fenugreek did x...."
Re: nutritional deficiency = physiological problems. I am not sure that these should be equated, as the latter typically implies anatomical or hormonal problems. Readers may mistakenly extrapolate the efficacy of fenugreek as being under all physiological conditions, not all of which were tested.
Re: 'milk flow'- what do you mean by this? Milk production, milk yield, or milk ejection/delivery? It seems that 'milk flow' is the preferred term when describing the deuterium model/results. But these terms appear to be used interchangeably at times and it can become confusing. It might be helpful to define milk flow at the start and differentiate its use from 'milk production' and 'milk yield.'
'lactation flux' - not a commonly used term, can you define it for the reader, please?
Line 44-46: the paragraph was about perceived insufficient milk production... then the sentence says 'Perception of insufficient milk secretion may result from many causes, including inability to lactate due to ........" that is not just a perception that is a REAL cause. It would be better to differentiated the perceived-- which is often taken as a misperception--and real problems.
Line 47: "Thus it is well established that mother milk production (incorrect grammar-- perhaps 'maternal milk production'?) often can be increased through psychological support or maternal breastfeeding education...." now the topic has suddenly swung back to misperceptions of low production, and it really isn't well established at all-- see this reference.
Murase, M., Wagner, E. A., C, J. C., Dewey, K. G., & Nommsen-Rivers, L. A. (2017). The Relation between Breast Milk Sodium to Potassium Ratio and Maternal Report of a Milk Supply Concern. J Pediatr, 181, 294-297.e293. doi:10.1016/j.jpeds.2016.10.044
Experimental design:
I had to draw my own flow chart to understand the distribution of the 72 rats, it was difficult to grasp quickly just from the narrative.
The one group that is missing from this picture is an 8 pups "control" (NP8F)- would have liked to see how fenugreek worked for that group versus the two challenge groups-- that would have made the comparisons complete.
Results:
One potential adverse effect of fenugreek that was not mentioned or explored is antithyroid activity. This has been demonstrated in rodent studies (Tahiliani, Panda), and there are an increasing number of anecdotal reports of fenugreek causing a drop in milk production in women who turn out to also be hypothyroid.
Author Response
Dear reviewer,
Thank you very much for reviewing our manuscript. We wanted to thank you for your complimentary and interesting comments and for your wording suggestions directly in the text. We have revised the manuscript according to your comments and we hope that you will find our responses satisfactory.
Please find attached a point-by-point response to your concerns.
Sincerly,
Thomas Sevrin

Reviewer 2 Report
Dear Authors,
Thank you for presenting a well constructed manuscript for review
Could you please mention the level of Fenugreek supplementation in the abstract Introduction could do with a discussion of the natural variation of galactologue effect in Fenugreek Your experimental design is complicated but is well explained and illustrated. One suggestion, and only a suggestion, is to label the groups in Figure 1 as the reference, control (fen high protein), Fenugreek high protein, control (fen low protein) and Fenugreek low protein or something similar to make linkage to the text easier for the reader Line 135 I believe the word should be killed rather than skilled Section 2.5. From your description of milk collection the macronutrient analysis could be confounded by the degree of fore,mid and hind milk collected. Could you please mention this in your macronutrient discussion (4.4) Section 2.8 I would like to see more clarity around the statistical analysis included please Tables 1, 2 and 3 correctly align treatment to the dam as that is your treatment unit as they are fed Fenugreek and the pup's growth etc are the main output measures. Table 4 appears to have moved away from this with n=22 for the NP:12, but the other groups seem to align with other Table,s n values with the dam as the treatment unit....so is this a typo? Section 4.4 first paragraph could do with further development and Table 2 certainly bears that out. This is a fascinating part of your outcomes - lactose concentration in the milk from NPF12 mothers has increased significantly over the NP:12 mothers but not the fat or protein. On a per day basis lactose is up 1.7 fold while the protein, fat and energy are only up 1.4 fold. Possibly the fenugreek supplementation is still not enough to account for 1.5 fold more pups Line 650 you'll see what to do If no patents were resulting (unlikely) you don't need this sectionAuthor Response
Dear reviewer,
Thank you very much for reviewing our manuscript. We wanted to thank you for your interesting comments and suggestions. We have revised the manuscript accordingly and we hope that you will find our responses satisfactory.
Please find attached a point-by-point response to your concerns.
Sincerly,
Thomas Sevrin
